# Phylogenetic analysis of *Mycobacterium bovis* reveals animal and zoonotic tuberculosis spread between Morocco and European countries

**Hind Yahyaoui Azami**[ID][1]\*, **Claudia Perea**[2], **Tod Stuber**[2], **Mohammed Bouslikhane**[3], **Jaouad Berrada**[3], **Hamid Aboukhassib**[4], **Alberto Oscar Allepuz Palau**[5], **Ana C. Reis**[6], **Mónica V. Cunha**[6,7], **Tyler C. Thacker**[2], **Suelee Robbe-Austerman**[2], **Liliana C. M. Salvador**[8], **Frederick D. Quinn**[ID][1]

**1** Department of Infectious Diseases, University of Georgia, Athens, Georgia, United States of America, **2** United States Department of Agriculture, Diagnostic Bacteriology and Pathology Laboratory, National Veterinary Services Laboratories, Animal and Plant Health Inspection Service, Ames, Iowa, United States of America, **3** Institut Agronomique et Veterinaire Hassan II, Rabat, Moroco, **4** Faculté Chouaib Doukkali, Avenue Jabran Khalil Jabran B.P, El Jadida, Morocco, **5** Departament de Sanitat i Anatomia Animals, Facultat de Veterinària, UAB, Barcelona, Spain, **6** cE3c- Center for Ecology, Evolution and Environmental Changes & CHANGE – Global Change and Sustainability Institute, Faculdade de Ciências da Universidade de Lisboa, Lisboa, Portugal, **7** BioISI- Biosystems and Integrative Sciences Institute, Faculdade de Ciências da Universidade de Lisboa, Lisboa, Portugal, **8** School of Animal and Comparative Biomedical Sciences, The University of Arizona, Tucson, Arizona, United States of America

\* Hind.YahyaouiAzami@uga.edu

## Abstract

Livestock production is a fundamental pillar of the Moroccan economy. Infectious diseases of cattle and other species represent a significant threat to the livestock industry, animal health, and food safety. Bovine tuberculosis (bTB), mainly caused by *Mycobacterium bovis*, generates considerable direct and indirect economic losses, and an underestimated human health burden caused by zoonotic transmission. Previous studies have suggested likely *M. bovis* transmission links between Morocco and Southern Europe, however, limitations inherent with the methods used prevented definitive conclusions. In this study, we employed whole genome sequencing analysis to determine the genetic diversity of the first 55 *M. bovis* whole-genomes in Morocco and to better define the phylogenetic links between strains from Morocco and a large dataset from related and neighboring countries. With a total of 780 *M. bovis* sequences extracted from cattle, wildlife or humans and representing 36 countries, we discovered two new *M bovis* spoligotypes in Morocco and that the Moroccan clonal complexes are classified as belonging to Europe or Unknown, supporting previous studies that the Sahara Desert might be playing a key role in preventing *M. bovis* transmission between North Africa and Sub-Saharan Africa. Furthermore, our analysis showed a close *M. bovis* genetic relationship between cattle from Morocco and cattle from Spain, France, Portugal and Germany, and from cattle in Morocco and humans in Italy, Germany, and the UK. These results suggest that animal trade and human migration between Morocco and these countries might be playing a role in disease transmission. Our study benefits from a large sample size and a rich dataset that includes

**Data availability statement:** All sequence data used for these analyses has been uploaded on to the National Centre for Biotechnology Information Short Read Archive (NCBI-SRA) under the Bioproject accession number PRJNA1200723. The individual sequences can be accessed under the following Biosample accession numbers: SAMN45911070 - SAMN45911124. All other metadata are available in the Supporting information files.

**Funding:** MVC acknowledges funding from Fundação para a Ciência e a Tecnologia (FCT)/ MCTES through national funds (PIDDAC) and co-funding from the European Regional Development Fund (FEDER) via the Lisbon Regional Operational Program and the Competitiveness and Internationalization Operational Program for Portugal 2020. This support is linked to the project "Colossus: Control Of Tuberculosis at the Wildlife/ Livestock Interface Using Innovative Nature-Based Solutions" (references PTDC/CVT-CVT/29783/2017, LISBOA-01-0145-FEDER-029783, POCI-01-0145-FEDER-029783). The funders played no role in study design, data collection, analysis, publication decisions, or manuscript preparation.

**Competing interests:** The authors have declared that no competing interests exist.

sequences from cattle, wildlife and humans from Morocco and neighboring countries, enabling the delineation of *M. bovis* genetic links across countries and host-species. Our study calls for further investigation of animal and zoonotic TB spread in Morocco and in other countries, which is important to inform future TB control measures at the animal-human interface.

## Author summary

Livestock production is a cornerstone of the Moroccan economy, but infectious diseases, particularly bovine tuberculosis (bTB), pose significant risks to animal health, and food safety. Predominantly caused by *Mycobacterium bovis*, bTB leads to substantial economic losses and an underestimated human health burden through zoonotic transmission. Previous research indicated possible *M. bovis* transmission links between Morocco and Southern Europe, though methodological limitations hindered conclusive findings. Here, we utilized whole genome sequencing to analyze the genetic diversity of the first 55 *M. bovis* genomes from Morocco, comparing them with a large dataset of 725 sequences from 36 countries, including cattle, wildlife, and human samples.

Our findings revealed two new *M. bovis* spoligotypes in Morocco and classified Moroccan clonal complexes as European or Unknown, suggesting the Sahara Desert limits transmission between North Africa and Sub-Saharan Africa. Additionally, we identified close genetic relationships between *M. bovis* from Morocco and strains from Spain, France, Portugal, Germany, and humans in Italy, Germany, and the UK, implying that animal trade and human migration might be contributing to disease spread.

This study, benefiting from a comprehensive dataset, underscores the need for further research on animal and zoonotic TB transmission to inform effective control measures at the animal-human interface.

## Introduction

Bovine tuberculosis (bTB) caused primarily by *Mycobacterium bovis (M. bovis)* is an important source of decreased animal health and economic distress in low- and middle- income countries. *M. bovis* can infect a wide range of hosts, including domestic livestock, wildlife, and humans; consequently, *M. bovis* may be an important and underappreciated human health burden in countries with infected livestock and wildlife reservoirs [1]. The human health burden of *M. bovis* is poorly investigated worldwide, and it can be underestimated, particularly in low-income countries where animal TB is endemic and the human access to health care is low. Cattle-to-human transmission of *M. bovis* is primarily caused by consumption of unpasteurized milk and milk products, and secondly by inhalation of aerosols through contact with infected cattle in the farm [2].

While animal TB has been eliminated from some high-income countries (e.g., Australia, Switzerland) [3,4], it remains an economic and health burden for several others where elimination efforts are hampered by spillovers from wildlife reservoirs into livestock herds and humans (e.g., USA [5,6], UK [7], Canada [8], New Zealand [9], Iberian Península [10,11]). Herd-to-herd transmission of *M. bovis* is variable depending on the geographic location. In

countries with low bTB prevalence and strong control efforts in place (e.g., frequent bTB testing, elimination of infected animals from the herds, abattoir surveillance, and monitoring/restriction of cattle movement), transmission between herds may be relatively low (e.g., 0.001% in the USA) [12]. However, despite strict test-and-slaughter programs, cattle movement can still drive bTB dissemination between herds [9]. In low- and middle-income countries, where livestock is traded without restrictions, *M. bovis* transmission between herds is common.

The last national bTB investigation in Morocco was conducted in 2003 [13]. In 2012, a pilot study took place in one of the provinces, it showed an individual prevalence of 20.4% (95% CI 18%–23%) and a herd prevalence of 57.7% (95% CI 48%–66%) [14]. Despite test and slaughter being the official control strategy for bTB in Morocco, it is not mandatory due to the high cost and lack of resources. Meat inspection at the slaughterhouse remains the only control strategy preventing infected meat from reaching consumers.

Phylogenetics of *M. bovis* associated with bTB cases have been increasingly investigated in the last 10 years [12,15]. Regions of deletion (RD) polymerase chain reaction (PCR), spoligotyping, and Mycobacterial Interspersed Repetitive Units- Variable Number of Tandem Repeats (MIRU-VNTR), in addition to other genotyping techniques, have been shown to be useful for *M. bovis* genetic analyses. When two or more techniques are used in conjunction, a higher genetic discrimination level between different *M. bovis* strains is achieved [16]. Moreover, with the recent increased use of whole genome sequencing (WGS) and single nucleotide polymorphism (SNP) analysis, a broader and more in-depth knowledge about bTB transmission dynamics within and between different countries and regions is being developed [17].

Based on genomic deletions (using genomic regions of difference 4 (RD4) and 9 (RD9)) and spoligotype, *Mycobacterium bovis* is classified into several groups of genotypes called clonal complexes (CC). African1 (AF1) [18], African 2 (AF2) [19], European1 (EU1) [20] and European2 (EU2) [21] are 4 major CCs of *M. bovis* that are well described in the literature.

In 2015, a molecular characterization of Moroccan *M. bovis* isolates from two slaughterhouses was performed, in which both deletion PCR and spoligotyping were used. From this study, SB0120, SB0121, and SB0265 were the most frequent spoligotypes found [22]. These have been previously reported in North African [23], European, and South and North American countries [24]. The results were supported by cattle trade data between those countries and Morocco, and the potential illegal importation of cattle from neighboring countries. Interestingly, the same study showed no spoligotypes matching to West, East, and Central Africa. The authors suggested that the desert acts like a barrier against the circulation of bTB between Sub Saharan African countries and Morocco [22]. However, deletion PCR and spoligotyping each target a single or very few genetic loci, covering less than 0.1% of the genome [25], therefore a higher resolution method is needed.

To this end, in this study we use a large number of whole-genome sequences and associated metadata to 1) characterize the genetic diversity of the first *M. bovis* WGS isolates from slaughtered cattle in Morocco and determine their clonal complexes; 2) compare the genetic diversity of *M. bovis* isolated from cattle in Morocco to *M. bovis* isolated from cattle, and wildlife in other different countries, and 3) identify signs of zoonotic transmission of *M. bovis* between Moroccan cattle and humans in other countries. We unraveled two new spoligotypes in Morocco and that Morocco's clonal complexes are more similar to the ones in Europe than the ones in Sub-Saharan Africa. Furthermore, we found close phylogenetic links between cattle in Morocco and cattle and humans in European countries, suggesting that animal trade between Morocco and these countries as well as human migration, are playing a role in the spread of TB.

## Materials and methods

### Ethical clearance

Gross visible lesions from slaughtered cattle were collected during routine meat inspection at slaughterhouses, therefore no ethical clearance was necessary.

### Sample collection

Gross visible lesions from slaughtered cattle were collected from three abattoirs located in Rabat, El Jadida, and Oujda (Fig 1). The collection period for Rabat was during March and April 2015 and for El Jadida was from June 2014 to July 2015. Samples collected from Oujda do not have information about the collection date. The length difference in sampling periods resulted in different numbers of isolates per location, where four were collected from Rabat, thirty-eight from El Jadida, and nine from Oujda (however, as mentioned above, no collection dates were available for this location). Four other *M. bovis* collected from cattle were added to the collection, but these had no metadata available. The difference of sampling period between Rabat and El Jadida was due to the lack of a country-wide surveillance program for bTB in Morocco and due to logistics related to each study site such as the availability of veterinaries to assist in the data collection and/or authorized transportation to the site. However, these

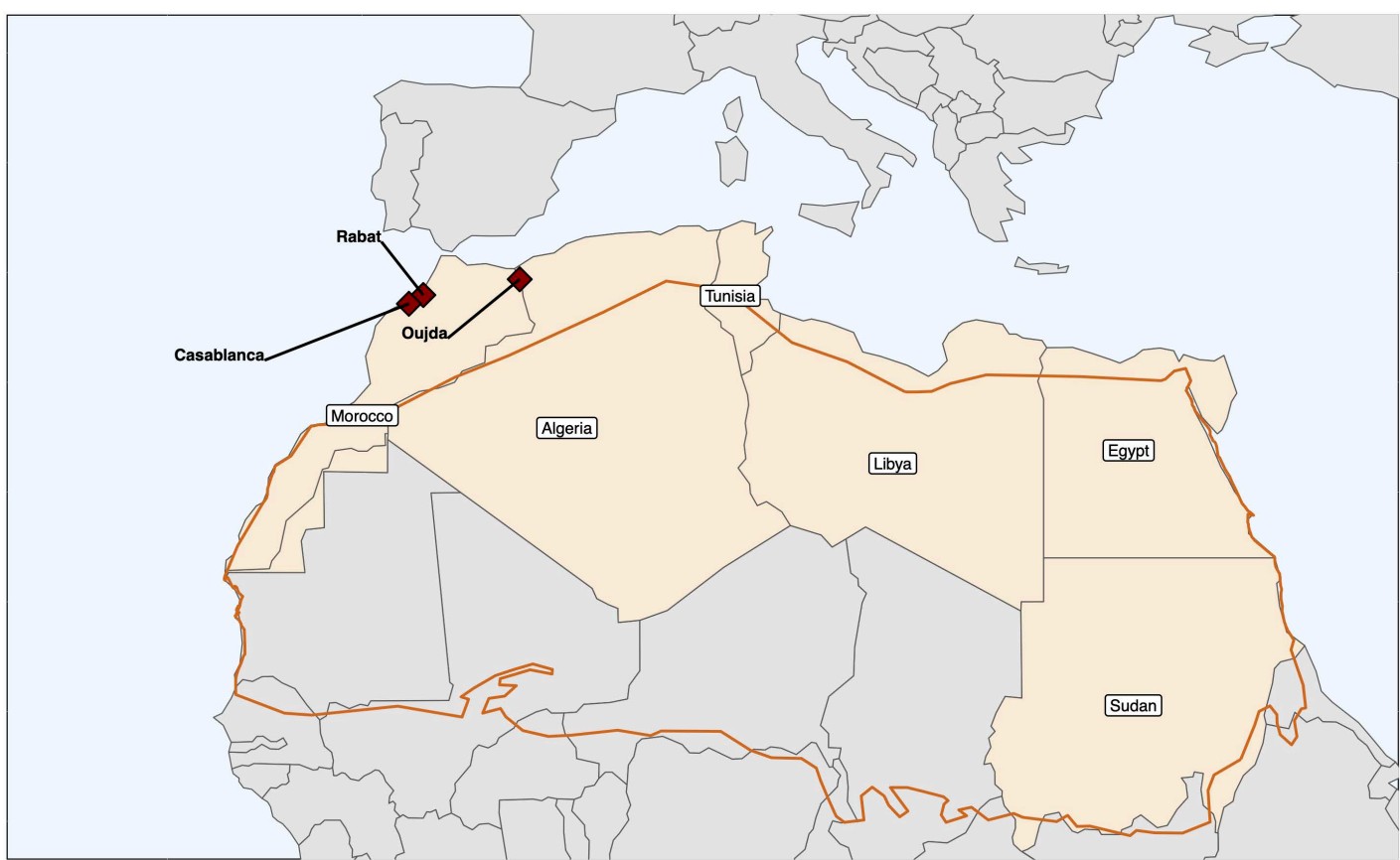

**Fig 1. Map showing the slaughterhouse locations from where tissue samples with evidence of bTB-like lesions were collected from cattle. The brown outline shows the Sahara Desert. We used the rnaturalearth package in R to download world country polygons from Natural Earth, sub-setting to Northern Africa. We produced the map using the sf and ggplot packages in R, plotting Oujda, Rabat, and Casablanca points using latitude and longitude data [46–48]. The data to build the map in Fig 1 came from here:** https://www.rdocumentation.org/packages/rnaturalearth/versions/1.0.1/topics/countries.

differences in sampling size are not important factor for our current study since we are not performing a comparison across study sites in Morocco, but a comparison between *M. bovis* samples from Morocco and samples from other countries. All Moroccon samples used in this study included lesions and granulomas suggestive of bTB found in the animals' tissues and associated lymph nodes.

The animals slaughtered at Rabat and El Jadida were mostly young bulls and older cows. These samples were composed of more male animals than female (n = 27 vs n = 15, out of 42). Most of these animals were crossbred (n = 36), while a few were labeled as imported breed (n = 6), including the most common imported cattle breed in Morocco, which is Holstein.

## Mycobacterial culture

Cattle with gross visible lesions suggesting bTB were sampled, and the lymph nodes and/or other tissues exhibiting lesions from each animal were collected, pooled and stored at −20°C until cultured. Tissue samples were processed in a biosafety level 3 laboratory using sterile surgical instruments.

Frozen samples were thawed overnight at 4°C. Subsequently, 5 g of lesion tissue and adjacent material was mixed with sterilized sand and 10 ml of red phenol. Aliquots of resulting solution (7.5 g) were placed in 15 ml conic tubes with 5 ml of 1N NaOH and incubated at room temperature for 10 minutes. HCl 6N was added for sample neutralization, and the tubes were centrifuged for 25 minutes at 3,000 X g.

The supernatants were discarded, and pellets were distributed in two pre-tested culture media: Lowenstein Jensen with pyruvate (LJP), and Herrold egg yolk medium. Cultures were incubated for 12 weeks at 37°C. Colonies were sub-cultured on 4 slants of LJP and incubated as previously described for the primary cultures. The cultures were examined weekly for assessment of growth.

## DNA purification

After 4 weeks of incubation at 37°C, the LJP slant cultures were heat inactivated following the USDA heat-killing protocol [17]. Briefly, the colonies from the four LJP slants were pooled, suspended in sterilized distilled water, and heated for 30 minutes at 100°C. After cooling to room temperature, the stocks were stored at −20°C.

The heat-killed mycobacteria were added to the prepared bead beater tubes with 1X Tris-EDTA (TE) buffer and phenol/chloroform/isoamyl alcohol (PCI), the samples were beaten at full speed for 2 minutes using a bead-beater machine (homogenizer). The samples were then spun at 16,000 X g for 5 minutes. The DNA contained in the top aqueous layer was removed (approximately 300 µl) and was added to a 1.5 ml microcentrifuge tube containing 30 µl of a 3M sodium acetate buffer solution. A total of 700 µl of 100% ice-cold ethanol was added to the samples. Samples were cooled at −80°C for 10–15 minutes, then spun at 4°C, 16,000 X g for 15 minutes. The ethanol was discarded and 1 ml ice-cold 70% ethanol was added. The tube was spun at 8,000 X g for 15 seconds and the remaining ethanol was removed. The samples were dried at room temperature and 300 µl of 1X TE buffer was added to resuspend the DNA, which was stored at 4°C for short term storage or −80°C for long term storage [26]. The DNA concentrations were measured using Nanodrop and Qubit 2.0 fluorometer according to manufacturer instructions and subsequently diluted to a starting concentration of 10 ng/µl.

## Whole genome sequencing

A minimum of 20 µL of DNA sample with a minimum concentration of 5 ng/µL was required for sequencing. Libraries were prepared using the Nextera XT Kit (Illumina, Inc., San Diego,

CA, USA), and sequencing was performed on an Illumina MiSeq device using 250 bp paired end read chemistry, according to manufacturer's instructions. Multiple isolates were indexed per lane, providing approximately 50–100X coverage per genome.

## Spoligotyping and clonal complexes

Spoligotype profiles were identified *in silico* using the WGS data used in this study (described below) and the National Veterinary Services Laboratory (NVSL) vSNP pipeline [27] with the "spoligo" function. The presence or absence of the spacer units in the mycobacterial genome are used to obtain the binary code, which is then cross-referenced against the *Mycobacterium bovis* Spoligotype Database (www.mbovis.org) to obtain the SB codes for each isolate. Clonal complex classification was based on previously published data [28], and on the most recent lineage classification [29].

## Phylogenetic analysis

We performed an analysis of the 55 collected isolates from Morocco for this study, and a large representative sample of publicly available isolates (725) (S1 File) downloaded from the NCBI Sequence Read Archive. A total of 780 *M. bovis* genomes were analyzed where thirty-six countries were represented (including Morocco; S1 File). The majority of the hosts associated with the analyzed genomes corresponded to cattle (n = 589), human (n = 107), and wildlife (several species, n = 76) (S1 File).

Raw FASTQ files were analyzed with the vSNP pipeline [30]. Briefly, the alignment against *M. bovis* AF2122/97 (NC_002945.4) was performed using Burrows-Wheeler Aligner [31] (BWA), and SNPs were called using Freebayes [32]. 80x dept of coverage was targeted. *Mycobacterium caprae* was used as the outgroup. Sites that fell within proline-glutamate (PE) and proline-proline-glutamate (PPE)-polymorphic CG-repetitive sequences (PGRS) were filtered and excluded, as well as SNP positions with a phred-scaled quality (QUAL) score for the alternate non-reference allele lower than 150 or when all positions in a data set had an allele count (AC) equal to 1 when analyzed as a diploid. Integrated Genomics Viewer [33] (IGV) was used to visually validate SNPs, and SNPs with mapping issues or alignment problems were manually filtered. A phylogenetic tree was constructed with RAxML [34] using a GTR-CAT model of substitution and a maximum-likelihood algorithm with the aligned whole-genome SNP sequences. Metadata such as 'Country' were color coded in the tree, however, only 7 countries were emphasized in the legend: in addition to Morocco, we included the countries from which Morocco imported cattle in the last 10 years (Spain, France, Germany, and Canada), and the countries that were geographically close to Morocco (Portugal and Algeria).

The output from the vSNP pipeline included: an alignment/fasta file containing the concatenated SNP sequences, an Excel table that includes the SNP position (with respect to the reference genome NC_002945.4), annotation and average mapping score of all the SNPs (which are grouped and sorted according to relatedness), and a phylogenetic tree. The accuracy of the phylogenetic tree was confirmed using the validated SNP table. Tree visualization, annotation, and editing were performed with FigTree [35] and iTOL [36].

## Results

### Spoligotypes of *M. bovis* from Morocco

Twenty-two spoligotyping patterns generated *in sillico* were identified for the Moroccan isolates (S2 File), of which two were new and were submitted to the https://www.mbovis.org/ database and assigned new SB numbers (SB2545 and SB2785). The most frequent patterns were SB0121 (n = 13, 23.2%), SB0265 (n = 12, 21.4%), and SB0120 (n = 8, 14.3%).

## *M. bovis* clonal complexes

The isolates analyzed in this study were assessed using the clonal complexes classification. In addition to the most recently described lineage classification, which identified further groups and defined some of the previously unknown groups. *Mycobacterium bovis* sequences from North Africa (Morocco and Algeria) were part of EU1, EU2, EU3, Unknown 1-PZAsus and Unknown 5 clonal complexes, while none of the isolates from North Africa were classified as AF1 and AF2 clonal complexes. All the isolates from AF1 and AF2 clonal complexes were from Sub-Saharan Africa except one (human *M. bovis* isolate collected in Switzerland) (Fig 2).

Most isolates were part of EU1, EU2, EU3-Unknown2 and AF1 clonal complexes, which are equivalent to La 8.1.8, La 1.7.1, La 1.2 and La 1.3 lineages, respectively, in the new clonal complexes' classification suggested by Zwyer et al [29].

The Moroccan isolates were part of eight genetic groups, based on NVSL classification [12]. Fig 3 shows the different groups highlighted in light grey and numbered from 1 to 8. Moroccan isolates are indicated with bold tree branches and human *M. bovis* are indicated with red marks (Fig 3).

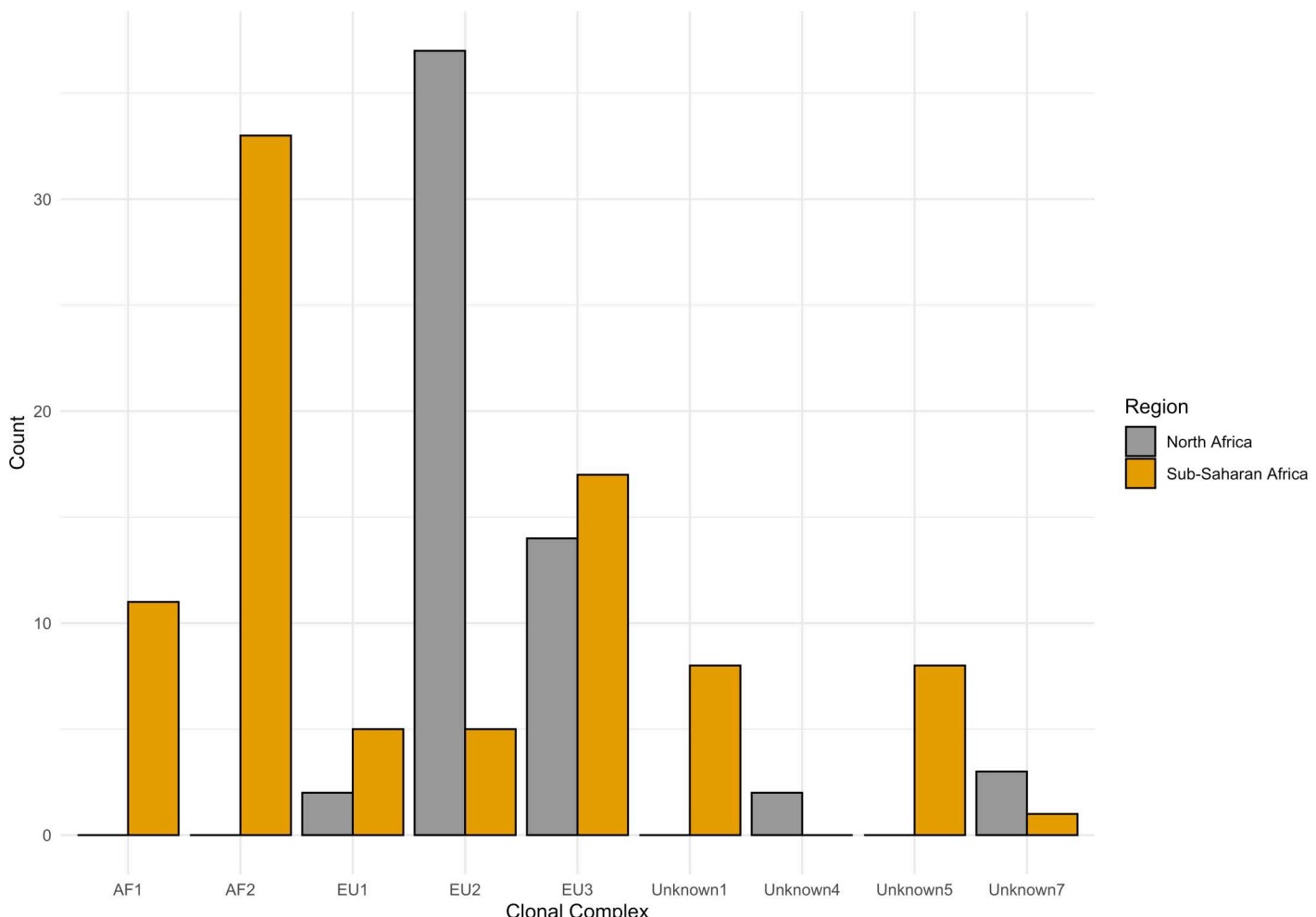

**Fig 2. Distribution of *Mycobacterium bovis* isolates from North Africa and Sub-Saharan Africa based on clonal complexes/lineages.**

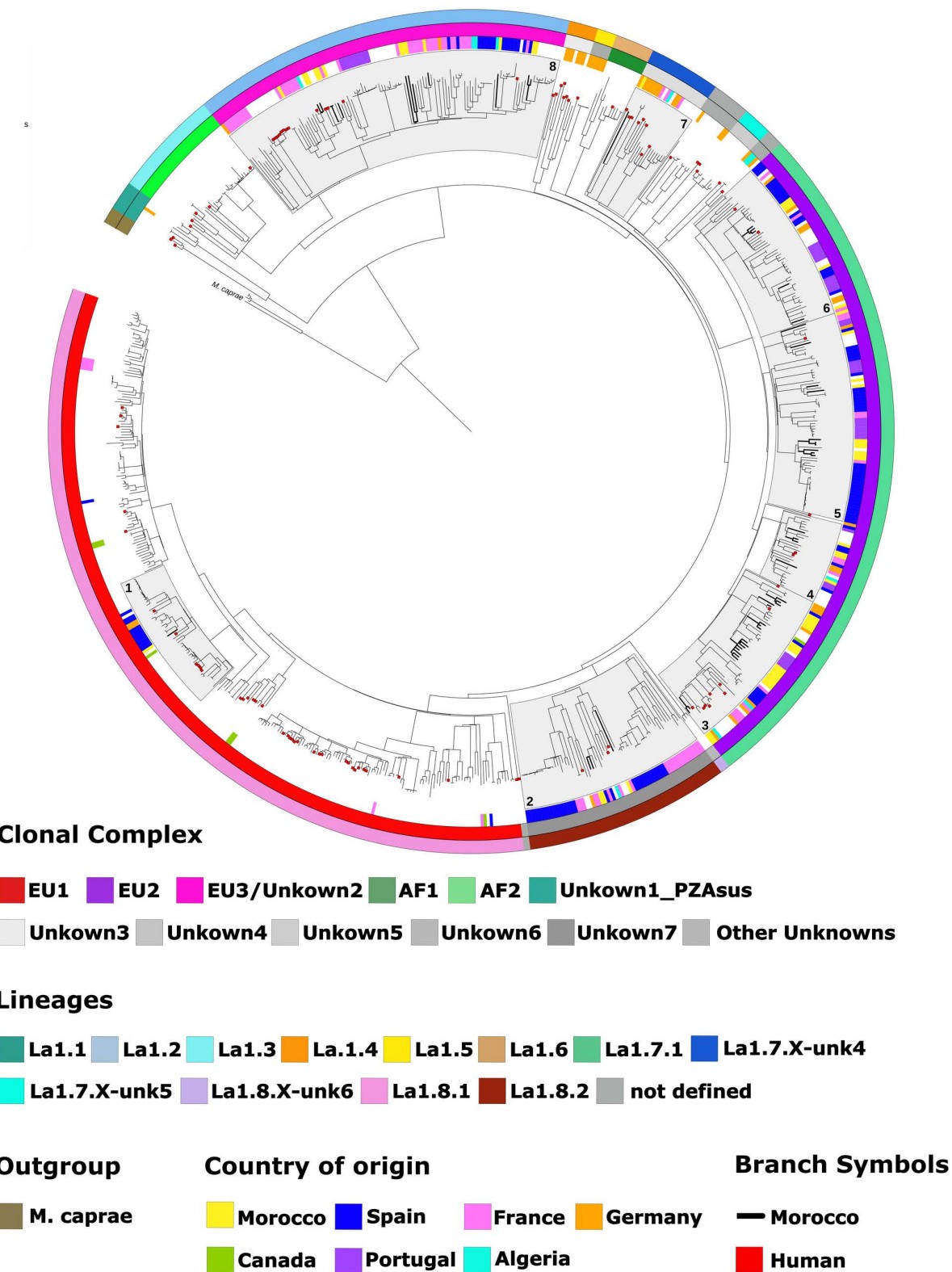

**Clonal Complex**

- EU1
- EU2
- EU3/Unkown2
- AF1
- AF2
- Unkown1_PZAsus
- Unkown3
- Unkown4
- Unkown5
- Unkown6
- Unkown7
- Other Unknowns

**Lineages**

- La1.1
- La1.2
- La1.3
- La.1.4
- La1.5
- La1.6
- La1.7.1
- La1.7.X-unk4
- La1.7.X-unk5
- La1.8.X-unk6
- La1.8.1
- La1.8.2
- not defined

**Outgroup**

- M. caprae

**Country of origin**

- Morocco
- Spain
- France
- Germany
- Canada
- Portugal
- Algeria

**Branch Symbols**

- Morocco
- Human

**Fig 3. Maximum-likelihood phylogenetic tree with all *Mycobacterium bovis* isolates used in this study.** The phylogenetic tree was generated using the GTR-CAT substitution model and presents all the analyzed *M. bovis* isolates used in the study (n = 725), with information about countries (inner circle), clonal complex (middle circle) and lineages (outer circle) color coded. Moroccan isolates are indicated with bold marked tree branches, and human isolates are indicated with a red dot. The areas filled with a light grey shade indicate the groups that include Moroccan isolates.

The SNP-based phylogenetic analysis performed on the 780 *M. bovis* isolates identified 8 genetic groups (Fig 4). Each of the eight genetic groups separately corresponding to groups 1 to 8 from Fig 3. Moroccan *M. bovis* sequences showed a variable genetic relatedness with isolates from the other emphasized countries. Fig 4A and 4D highlight the genetic relationship between *M. bovis* isolates from Morocco and Spain at different degrees, sharing a most recent common ancestor (MRCA) with ≥30 SNPs. In addition, *M. bovis* isolates from Morocco and France also showed a tendency to cluster together, sharing a MRCA with ≥50 SNP (Fig 4E, 4F, and 4H). Only 7 *M. bovis* from Canada were publicly available at NCBI and included in the analysis, with only 2 falling into the same groups as the Moroccan isolates (Fig 4D, 4A, and 4C), however, no close relationships were noted (>100 SNP to a MRCA). Moroccan and Portuguese *M. bovis* isolates showed a close genetic relationship, sharing a MRCA within 50 to 120 SNPs (Fig 4D, 4C, and 4E). Finally, two *M. bovis* isolates from Morocco shared a MRCA with an isolate from Algeria with 50 SNPs (Fig 4D and 4C).

The database analyzed here included 107 *M. bovis* isolates from human patients from Germany, USA, Mexico, UK and Italy. The closest matches of Moroccan cattle to human isolates were from Germany (Fig 4D, 4C and 4G), and Italy (Fig 4D and 4H), having shared a MRCA with ≥20 and ≥16 SNPs, respectively.

## Discussion

The current study represents the first in depth WGS phylogenetic study and SNP analysis of *M. bovis* isolates in Morocco. The analysis included 55 *M. bovis* isolates from three different slaughterhouses in the north of the country and 725 *M. bovis* whole genome sequences downloaded from the NCBI sequence read archive. The results showed that isolates from Morocco are genetically related to isolates from Spain, France, and at a lesser level to isolates from Portugal, Algeria, and Canada. The genetic similarity of M. bovis isolates between Morocco and the listed countries is reinforced by Morocco's robust economic and historical connections with Spain, France, and Portugal, alongside its decade-long importation of cattle from France, Spain, Canada, and Germany.

### Spoligotypes and clonal complexes

The spoligotypes patterns generated *in sillico in the present study were a total of 22, with two new spoligotypes (*SB2545 and SB2785). The most frequent patterns were SB0121 (n = 13, 23.2%), SB0265 (n = 12, 21.4%), and SB0120 (n = 8, 14.3%), which is similar to previous findings in Morocco [22].

AF1 and AF2 have been shown to be limited to Africa, while EU1 and EU2 have been described in Europe and South America [16]. North Africa has a unique geographical location; it is separated from Sub-Saharan Africa by the Sahara Desert, and it has proximity to Europe, which facilitates commercial trade and human migration. In the current study, *M. bovis* isolates from North Africa were classified as EU1 and EU2 CCs, and none of them fell under AF1 or AF2. This suggests that the Sahara Desert might play a role as a geographical buffer preventing circulation of *M. bovis* between North Africa and Sub-Saharan Africa. This is in line with previous hypothesis based on spoligotype findings in Morocco [22].

### *M. bovis* genetic diversity: Cattle and wildlife

The SNP analysis shows *M. bovis* strain similarity between the different countries suggesting that there was a migration of strains perhaps through a combination of human migration (from Morocco to Europe) and cattle movement (from Europe to Morocco). Data from other countries, which are historically and economically linked to Morocco, have not been available.

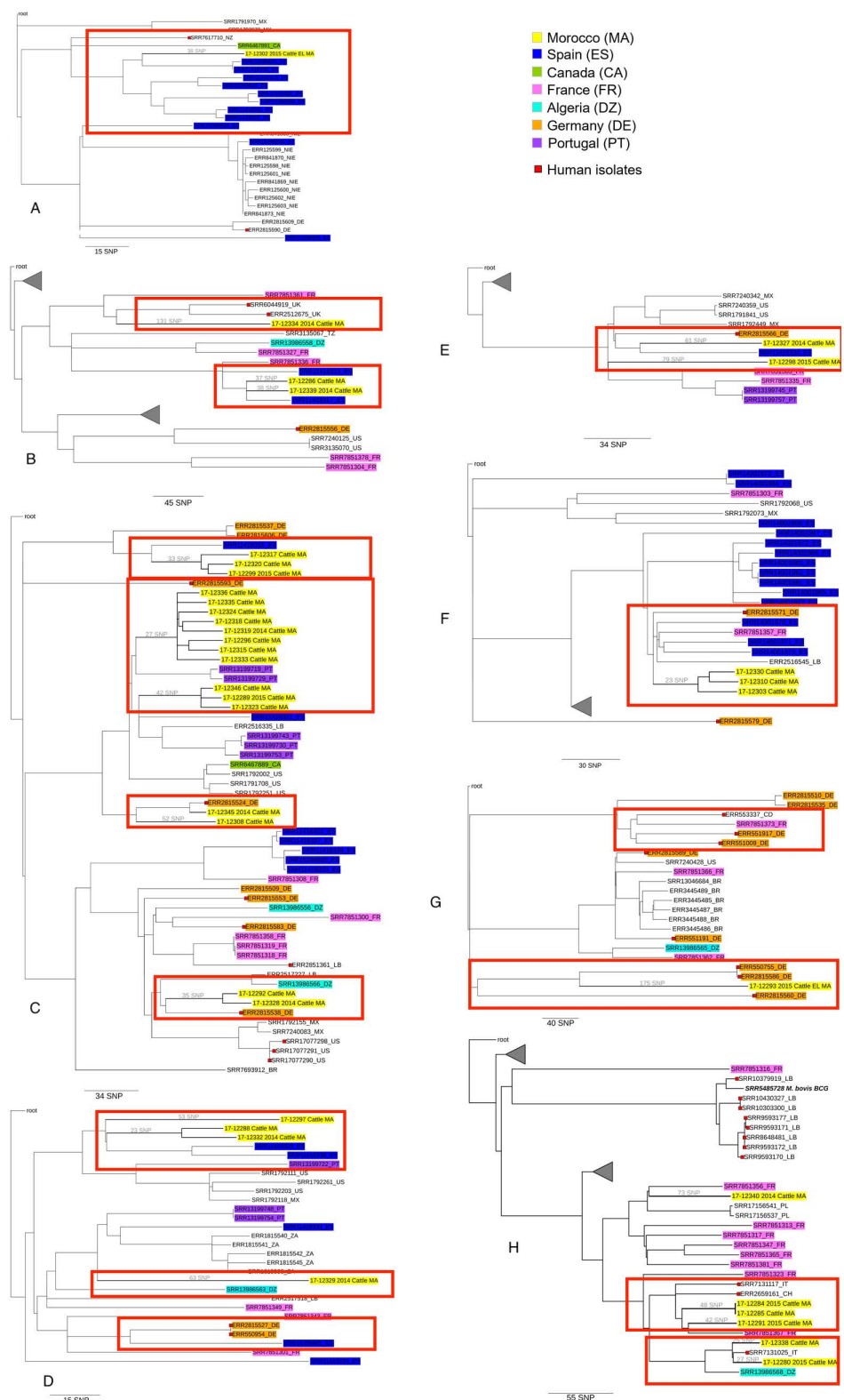

**Fig 4. High-resolution maximum-likelihood phylogenetic tree of groups 1–8.** The legend indicates country of origin based on the highlighted isolate color. The scale bar represents the branch length in SNPs. Within this figure, eight panels illustrate the genetic relationships among *M. bovis* isolates. Fig 4A highlights the genetic connection between Moroccan and Spanish *M. bovis* isolates from cattle, alongside one of the seven Canadian *M. bovis* isolates

                 *Mycobacterium bovis* phylogenetic spread between Morocco and Europe

analyzed. Fig 4B reveals genetic relationships between *M. bovis* isolates from Moroccan cattle and UK human in the top red box, while the bottom red box showcases the close genetic proximity of *M. bovis* isolates between Moroccan and Spanish cattle. Fig 4C emphasizes genetic proximity between *M. bovis* isolates from Moroccan and Spanish cattle in the top red box, with the second box displaying clustering of *M. bovis* isolates between Moroccan and Portuguese cattle. The third and fourth red boxes in Panel 3 show that *M. bovis* from Moroccan cattle are genetically related with *M. bovis* isolates collected from humans in Germany (including one that shares their Most Recent Common Ancestor with 20 SNPs apart). The bottom box also highlights another closely related relationship between a *M. bovis* isolate from Moroccan cattle a *M. bovis* isolate from human in Germany, as well as between *M. bovis* in Moroccan cattle and *M. bovis* in Algerian cattle. Fig 4D demonstrates the close genetic relationship between *M. bovis* cattle isolates from Morocco, Spain, and Portugal. Fig 4E displays genetic relationships between *M. bovis* isolates from Moroccan cattle and Spanish cattle and Moroccan cattle and humans in Germany. Fig 4F does not show close genetic links between *M. bovis* isolates from Morrocan cattle and cattle or humans from other countries. Fig 4G shows a close genetic relationship between *M. bovis* from Moroccan cattle and three humans from Germany. Fig 4H's red box illustrates the genetic clustering of *M. bovis* isolates between cattle from Morocco and humans from Italy and Switzerland. The bottom red box highlights the close genetic relationship of *M. bovis* isolates between cattle in Morocco and a human in Italy, as well as the close relationship of these isolates with one from cattle in Algeria.

However, we hypothesize that similar links to those observed between Morocco and Spain will exist between Morocco and other European countries, such as Belgium, Germany, Netherlands, Austria and Ireland, from where Morocco has imported cattle for breeding in the past [37]. The results of the present study showed no genetic links between M. bovis isolates from cattle from Morocco and wildlife from other countries. However, *Mycobacterium bovis* has been identified among 6 wild boars in Morocco among 43 tested animals [38]; nevertheless, no data are available for the interaction between cattle and wild boar, and the transmission dynamics between the two species. It should be highlighted that wild boar hunting is practiced in Morocco, resulting in human exposure to *M. bovis* and potential transmission from wild boar to humans.

## *M. bovis* genetic diversity: cattle to human

The analysis included 107 human *M. bovis* isolates from different countries, and genetic similarities were identified between Moroccan cattle *M. bovis* isolates and human *M. bovis* from Italy, Germany, and the UK. During World War I, migration from Morocco to Europe started and it has increased throughout the years, currently there are approximately 5 million Moroccans living abroad, mostly in Europe [39]. This human movement from Morocco to European countries can explain the genetic similarities between cattle *M. bovis* strains from Morocco and human *M. bovis* strains from Europe. Transmission of *M. bovis* from cattle to humans has been documented previously in several countries. A study in Mexico showed that 30.2% of 533 human tuberculosis patients were infected with *M. bovis* [40]. In Tunisia, two separate studies showed that extrapulmonary TB (EPTB) was caused mainly by *M. bovis* (76% and 77%, respectively) [23,41]. Spoligotyping and MIRU-VNTR analyses of isolates from EPTB patients showed a similarity between the strains isolated from humans and cattle; in addition, most of *M. bovis* TB patients have reported close contact with livestock and consumption of unpasteurized milk and milk products [42]. Zoonotic TB has not been investigated in Morocco, however, bTB has a high individual (20.4%) and herd (57.7%) prevalence in cattle [43], and livestock keepers and their families are in very close contact with cattle. These, together with the absence of an official milk pasteurization policy in Morocco, support the likelihood of a high risk of *M. bovis* transmission to humans, particularly in the agriculture sectors of Morocco.

Taking into consideration that bTB is endemic in Moroccan cattle, while the zoonotic human health burden is still unknown, the inspection/condemnation method puts slaughterhouse workers, as well as the general human health, at risk of contracting *M. bovis*. In

addition, tracking the animals to their herds is not possible because no adequate tracing systems are in place. Having a detailed movement tracking of cattle is an important step in the control of any infectious disease, and it will make a potential test and slaughter intervention to control bTB easier to perform, and more sustainable. Morocco was in the process of launching a campaign to ear tag all the cattle in the country, however, those efforts came to a halt because of the COVID-19 pandemic.

The gold standard for bTB control is test and slaughter, which is based on testing cattle using the simple and comparative tuberculin skin tests followed by the slaughter of test positive animals. Sensitivity to the simple and comparative tuberculin skin tests is not optimal. A low sensitivity will lead to a low detection of infected animals, allowing the persistence of transmitters within the population and leading to a longer time to elimination of the disease. In addition, a mathematical transmission model for bTB in Morocco has shown that the key element regarding bTB elimination is sensitivity of the diagnostic test used [44]. The addition of WGS, when applied to a larger sample size in different areas of Morocco, coupled with strong epidemiological data, can lead to the identification of high transmission areas and potentially sources of transmission in those areas. The application of another screening test in high transmission areas, such as the IFN gamma response assay, which has been shown to have enhanced sensitivity when combined to tuberculin skin test, will help to decrease the false negative results [45]. Lastly, for an optimal bTB control plan, *M. bovis* infection should be investigated in other susceptible hosts within the geographic area, in the case of Morocco, humans, wild boar, small ruminants and camels. Moreover, comparing additional Moroccan isolates with European and African animal and human isolates will provide additional insight regarding the transmission dynamics of *M. bovis* between Europe and Africa, and will aid in preventing future dissemination and re-introductions of the disease.

The present study included 55 sequences of *M. bovis* collected from slaughtered cattle in Morocco. Analyzing these samples alongside 725 other *M. bovis* sequences from relevant countries provided insights into the spread of *M. bovis* between Africa and Europe. However, a much larger number of *M. bovis* isolates is needed from Morocco to test the strength of the genetic links between *M. bovis* from Morocco and Europe. One of the limitations of the study is that the duration of collection of Moroccan *M. bovis* samples included in the analysis was different across the collection sites, which could have influenced the overall genetic diversity found in Morocco, and therefore, not be representative of the full genetic diversity of *M. bovis* in the country and its relatedness to neighboring countries. However, in our study, we analyzed the samples from the different collection sites together and compared them with samples from other countries, therefore, the possible sampling bias across sampling locations does not change the integrity of our results that show *M. bovis* phylogenetic links between cattle in Morocco and cattle and humans in European countries. It is important to notice that the genetic diversity found in our samples is not representative of the genetic diversity and clonal complexes existing in Morocco. Our study found two new spoligotype patterns in Morocco, but there might be more. As such, a more systematic sampling over a longer period of time is needed to be able to capture the overall genetic diversity and clonal complexes in Morocco. Additionally, the human samples included in the analysis lacked attached metadata, making it difficult to draw any definitive epidemiological conclusions about *M. bovis* cattle to human transmission. To determine the exact burden of *M. bovis* in animal and human health in Morocco, a significant sample size of human and cattle *M. bovis* isolates should be collected from the same region over a one to two years period following the One Health approach. Samples from wildlife will also offer another layer of understanding of *M. bovis* transmission dynamics. Whole genome sequencing will be then an excellent tool to use in order to determine transmission patterns of *M. bovis* in the cattle-wildlife-human interface in Morocco.

## Conclusion

In conclusion, this study provides the first in depth whole genome sequencing (WGS) phylogenetic analysis and SNP analysis of *M. bovis* isolates in Morocco, revealing significant genetic links between Moroccan and European isolates. The study highlights the need for improved cattle tracking systems and enhanced diagnostic methods to effectively control bovine tuberculosis (bTB) in Morocco. Expanding WGS analysis to include a broader range of samples and hosts, along with implementing a comprehensive bTB control plan, will be crucial in mitigating the disease's impact on both animal and human health in the region.

## Supporting information

**S1 File. Metadata overview for *Mycobacterium bovis* samples analyzed for the present study, including Sequence ID, host species, United States Department of Agriculture - National Veterinary Services Laboratory *M. bovis* worldwide metadata project classification, clonal complex, lineage, and two-letter country code.** This table highlights the genetic diversity and geographic origins of the samples included in the study.
(XLSX)

**S2 File. Detailed information for Moroccan *Mycobacterium bovis* isolates, including sample identifiers, sequence references, sequencing statistics, and spoligotyping codes.** This data provides insights into the genetic diversity and spoligotype patterns of *M. bovis* strains isolated in Morocco.
(XLSX)

## Acknowledgment

We would like to acknowledge Gabriella Veytsel for helping with Fig 1 generation, Alexander Bucksch for helping with Fig 4 visualization, and Nicholas Foster for uploading the fastq sequences to NCBI-SRA. Strategic funding from FCT to cE3c and BioISI Research Units (UIDB/00329/2020 and UIDB/04046/2020, respectively) and to the associate laboratory CHANGE (LA/P/0121/2020) are gratefully acknowledged.

## Author contributions

**Conceptualization:** Hind Yahyaoui Azami, Claudia Perea, Tod Stuber, Liliana C M Salvador, Frederick D. Quinn.

**Data curation:** Hind Yahyaoui Azami, Claudia Perea, Tod Stuber, Mohammed Bouslikhane, Jaouad Berrada, Hamid Aboukhassib, Alberto Oscar Allepuz Palau, Ana C Reis, Mónica V Cunha, Tyler C Thacker, Suelee Robbe-Austerman, Liliana C M Salvador, Frederick D. Quinn.

**Supervision:** Suelee Robbe-Austerman, Liliana C M Salvador, Frederick D. Quinn.

**Writing – original draft:** Hind Yahyaoui Azami, Claudia Perea, Tod Stuber, Liliana C M Salvador, Frederick D. Quinn.

**Writing – review & editing:** Hind Yahyaoui Azami, Claudia Perea, Tod Stuber, Ana C Reis, Mónica V Cunha, Tyler C Thacker, Suelee Robbe-Austerman, Liliana C M Salvador, Frederick D. Quinn.

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
