## [Decision Letter · Decision Letter 0]

25 Mar 2024

Dear Dr. Yahyaoui Azami,

Thank you very much for submitting your manuscript "Phylogenetic analysis of Mycobacterium bovis Reveals Evidence Of Animal And Zoonotic Tuberculosis Transmission Between Morocco And European Countries" for consideration at PLOS Neglected Tropical Diseases. As with all papers reviewed by the journal, your manuscript was reviewed by members of the editorial board and by several independent reviewers. In light of the reviews (below this email), we would like to invite the resubmission of a significantly-revised version that takes into account the reviewers' comments. 

We cannot make any decision about publication until we have seen the revised manuscript and your response to the reviewers' comments. Your revised manuscript is also likely to be sent to reviewers for further evaluation.

Sincerely,

Elsio A Wunder Jr, DVM, Ph.D.

Section Editor

Elsio Wunder Jr

Section Editor

Reviewer's Responses to Questions

**Key Review Criteria Required for Acceptance?**

**Methods**

-Are the objectives of the study clearly articulated with a clear testable hypothesis stated?

-Is the study design appropriate to address the stated objectives?

-Is the population clearly described and appropriate for the hypothesis being tested?

-Is the sample size sufficient to ensure adequate power to address the hypothesis being tested?

-Were correct statistical analysis used to support conclusions?

-Are there concerns about ethical or regulatory requirements being met?

Reviewer #1: 1) objectives are clear but the article has deviated from main objectives.

2) The study design is ok.

3) Population is not clearly defined.

4) No issues with sample size.

5) No ethical issues.

Reviewer #2: 1. Objectives "to determine the genetic relatedness and epidemiological links" comes late in the Introduction, and should be moved to the opening paragraphs. this focuses the reader better. The introduction is too long and could be restructured to explain: i) the impact of M bovis in Morocco for animal health (with limited economic data from nearby Ethiopia) ii) the potential for M bovis to be spread to humans and wildlife iii) initial molecular analysis using older methods iv) the role of WGS in clarifying genetic relationships

2. Study design is adequate, but vulnerable with the highly variable comparison data from isolates outside Morocco. The design might be reformatted to present Moroccan cattle isolates compared to a: other cattle, b: human isolates, c: wildlife?

3. The population is clearly described.

4. Sample size is adequate.

5. Statistical analysis is adequate.

6. No concerns for ethical or regulatory requirements.

**Results**

-Does the analysis presented match the analysis plan?

-Are the results clearly and completely presented?

-Are the figures (Tables, Images) of sufficient quality for clarity?

Reviewer #1: The results are not clearly presented and the reader may find it challenging to get to the main point.

The tables are ok but the dataset is not analyzed adequately.

Reviewer #2: 1. analysis matches the plan

2. the results are challenging to present, since the Moroccan isolates and the reference isolates are so different.

3. TAble 1 does not help the manuscript, and could be summarized in a paragraph.

Figure 1 map is OK, but adding the location of Sahara Desert would be helpful. 12 of 52 animals were imported from where? could importation have any relevance to their genetic analysis?

Figure 2 is unreadable. Only the outer circle has numbers; zooming in only shows blurred characters. This Figure should be removed.

Figure 3 contains important data, but is almost too much for the reader to comprehend. 

Figure 4 contains the most important data the authors are presenting. This should be made into 7 or 8 individual Figures, so that the Discussion can address each Panel properly and separately. Group 6 has no comments, and could be excluded?

Figure 5 is tangential data, and does not contribute to the article.

The WGS analysis is easier to comprehend than the lineage and clonal complex data.

**Conclusions**

-Are the conclusions supported by the data presented?

-Are the limitations of analysis clearly described?

-Do the authors discuss how these data can be helpful to advance our understanding of the topic under study?

-Is public health relevance addressed?

Reviewer #1: No conclusion is present. 

Limitations and strengths of the study are not reported. It is also not mentioned what knowledge or implementation gap exactly they have addressed.

Reviewer #2: The conclusions show the strength of WGS analysis in revealing connections among isolates from a wide variety of sources. Again the conclusions could be segregated into sections of a: cattle to cattle comparisons, b: cattle to human, and c: cattle to wildlife. 

This is an initial attempt at WGS analysis, and the comparison data set is so different making comparisons challenging.

Public health relevance is paramount for this kind of manuscript, and needs considerably more emphasis of both the animal and human health implications.

**Editorial and Data Presentation Modifications?**

Reviewer #1: Minor comments:

1) line 36: “and food safety and security.” No need for two “ands”

2) Introduction, line 63: “decreased animal health and economic distress in many countries of the world”, the term “many countries” is not specific, I would suggest to be more specific on it. 

3) Sample collection is not clearly explained in terms of choosing the sample. Was the examiner looking for all organs and inspecting the organs separately or only the animals’ lungs?

4) Table-1, “Areas of origin” it is not clearly understandable if the mentioned areas are the where animals were imported? Or Slaughtered? 

5) For mycobacterial culture and genome sequencing part: No references are provided. This makes it unclear whether you followed WHO/CDC guidelines or any other SOPs.

6) EU-1, EU-2, EU-3 and AF1 are poorly explained and reader may not understand clearly about it.

7) Table-1: you have provided data about 52 samples from different parts of Morocco but in discussion you have mentioned 56 isolated. 

8) In the sample collection you have mentioned “Rabat, El Jadida (western Morocco) and Oujda (Eastern Morocco)” but in discussion you have related them all to the north of Morocco. The reason for choosing the north of Morocco is also poorly explained in the text.

9) line 340-341: you have mentioned that the closest match of Moroccan cattle to human isolates were from Germany, UK and Italy but in the discussion part, you have stated that: the results showed that isolates from Morocco are genetically related to isolates from Spain, France. How would you explain this?

10) line 80: based on the reference you provided it has mentioned “North America” not the USA.

11) line 86: it is important to mention at what number or proportion it has been isolated from Eurasian wild boar. 

12) line 93 is in opposition with line 80, see comment 5.

13) line 96: “2012 in Morocco, bTB herd prevalence was 56%” based on your reference the percentages are different with what you have reported. “20.4% (95% CI 18%-23%) and 57.7% (95% CI 48%-66%), respectively, were observed in this study.”

14) line 109-117: The reference for this paragraph is missing however, above these paragraph 5 references are given for a single sentence. (line 108-109)

15) line 130: space is needed after “[19]”.

16) Line 183: “Cattle with gross visible lesions suggesting bTB were sampled”. Based on what was explained earlier the examiner has visited the slaughterhouse and taking sample from the slaughtered animals’ organs. This is not in aligned with your sampling.

17) line 254: In total 780 genomes were analyzed but in line 273 it says 784. Is there any thing missing?

Reviewer #2: (No Response)

**Summary and General Comments**

Reviewer #1: Major comments:

1) Line 36: “Bovine tuberculosis (bTB), mainly caused by Mycobacterium bovis (M. bovis)” the sentence gives the impression that there are also some other causes for bTB also. 

2) Line 38: You have mentioned “in addition to the unknown human health burden caused by zoonotic transmission” I would suggest to go through literature again. I have provided an example: https://pubmed.ncbi.nlm.nih.gov/27697390/

3) Line 54: The term “North Africa” also includes Morocco. How would you justify the sentence “no M. bovis sequences from North Africa in the present database were classified”????

4) The data that which is provided in the supplements is poorly analyzed.

5) In sample collection it is not clearly explained that, why would sample collection duration vary between regions?

6) Line 71: “…….contact with animal carcasses and infected organs at slaughter houses” based on the reference that you provided it has reported: “2101 cases of pulmonary TB from the Cheshire sanatorium represented the period 1934–1943. Of these, just 48 cases (2.28%) were bTB and only 10 were strongly suggestive of airborne transmission from cattle”. This is not aligned with what you have written.

7) line 72: The comparison of bTB cost between Morocco and Ethiopia is not quite convincing, how would you justify this comparison?

8) The discussion part is not well organized and does not present a clear message. There are many controversies in the discussion part which are not clearly explained. 

9) What are the strengths and limitations of this study? It has been known for decades that mycobacterium bovis can be transmitted from animals such as bovine to humans, what knowledge or implementation gap you have addressed in this manuscript?

10) There is no conclusion part to summarize the findings and gaps for future research.

Reviewer #2: The strength of this paper is the enormous amount of molecular data has been obtained for analysis. But the execution is somewhat rambling and would benefit from reformatting the Introduction, Results and Discussion into subsections of cattle - cattle; cattle - human and cattle - wildlife datasets.

PLOS authors have the option to publish the peer review history of their article (what does this mean? ). If published, this will include your full peer review and any attached files.

**Do you want your identity to be public for this peer review?** For information about this choice, including consent withdrawal, please see our Privacy Policy .

Reviewer #1: Yes: Dr. Ahmad Reza Yosofi MD, M.Sc.

Reviewer #2: No
---

## [Decision Letter · Decision Letter 1]

25 Sep 2024

Dear Dr. Yahyaoui Azami,

Thank you very much for submitting your manuscript "Phylogenetic analysis of Mycobacterium bovis reveals animal and zoonotic tuberculosis transmission between Morocco and European countries" for consideration at PLOS Neglected Tropical Diseases. As with all papers reviewed by the journal, your manuscript was reviewed by members of the editorial board and by several independent reviewers. The reviewers appreciated the attention to an important topic. Based on the reviews, we are likely to accept this manuscript for publication, providing that you modify the manuscript according to the review recommendations. 

Sincerely,

Elsio A Wunder Jr, DVM, Ph.D.

Section Editor

Elsio Wunder Jr

Section Editor

Reviewer's Responses to Questions

**Key Review Criteria Required for Acceptance?**

**Methods**

-Are the objectives of the study clearly articulated with a clear testable hypothesis stated?

-Is the study design appropriate to address the stated objectives?

-Is the population clearly described and appropriate for the hypothesis being tested?

-Is the sample size sufficient to ensure adequate power to address the hypothesis being tested?

-Were correct statistical analysis used to support conclusions?

-Are there concerns about ethical or regulatory requirements being met?

Reviewer #1: yes

**Results**

-Does the analysis presented match the analysis plan?

-Are the results clearly and completely presented?

-Are the figures (Tables, Images) of sufficient quality for clarity?

Reviewer #1: Figure 3 and figure 4 quality are very poor. i would recommend to replace them with a better quality images.

**Conclusions**

-Are the conclusions supported by the data presented?

-Are the limitations of analysis clearly described?

-Do the authors discuss how these data can be helpful to advance our understanding of the topic under study?

-Is public health relevance addressed?

Reviewer #1: Yes

**Editorial and Data Presentation Modifications?**

Reviewer #1: (No Response)

**Summary and General Comments**

Reviewer #1: (No Response)

PLOS authors have the option to publish the peer review history of their article (what does this mean? ). If published, this will include your full peer review and any attached files.

**Do you want your identity to be public for this peer review?** For information about this choice, including consent withdrawal, please see our Privacy Policy .

Reviewer #1: Yes: Ahmad Reza Yosofi MD, M.Sc.

Figure Files:

Data Requirements:

Reproducibility:

References

---

## [Editor Report · Decision Letter 2]

24 Dec 2024

Dear Dr. Yahyaoui Azami,

We are pleased to inform you that your manuscript 'Phylogenetic analysis of Mycobacterium bovis reveals animal and zoonotic tuberculosis spread between Morocco and European countries' has been provisionally accepted for publication in PLOS Neglected Tropical Diseases.

Best regards,

Elsio A Wunder Jr, DVM, Ph.D.

Section Editor

Elsio Wunder Jr

Section Editor

Shaden Kamhawi

co-Editor-in-Chief

Paul Brindley

co-Editor-in-Chief

---

## [Editor Report · Acceptance letter]

Dear Dr. Yahyaoui Azami,

We are delighted to inform you that your manuscript, "Phylogenetic analysis of Mycobacterium bovis reveals animal and zoonotic tuberculosis spread between Morocco and European countries," has been formally accepted for publication in PLOS Neglected Tropical Diseases.

Best regards,

Shaden Kamhawi

co-Editor-in-Chief

Paul Brindley

co-Editor-in-Chief
